# Chickpea-Derived Prebiotic Substances Trigger Biofilm Formation by *Bacillus subtilis*

**DOI:** 10.3390/nu13124228

**Published:** 2021-11-25

**Authors:** Yaa Serwaah Amoah, Satish Kumar Rajasekharan, Ram Reifen, Moshe Shemesh

**Affiliations:** 1Department of Food Sciences, Institute for Postharvest Technology and Food Sciences, Agricultural Research Organization (ARO), Volcani Institute, Rishon LeZion 7528809, Israel; yaa.amoah@mail.huji.ac.il (Y.S.A.); generic.sat@gmail.com (S.K.R.); 2The Robert H. Smith Faculty of Agriculture, Food and Environment, The Hebrew University of Jerusalem, Rehovot 7610001, Israel; ram.reifen@mail.huji.ac.il

**Keywords:** probiotic bacteria, chickpea fiber, beneficial biofilm, synbiotic food, functional probiotics

## Abstract

Chickpea-based foods are known for their low allergenicity and rich nutritional package. As an essential dietary legume, chickpea is often processed into milk or hummus or as an industrial source of protein and starch. The current study explores the feasibility of using the chickpea-derived prebiotic substances as a scaffold for growing *Bacillus subtilis* (a prospective probiotic bacterium) to develop synbiotic chickpea-based functional food. We report that the chickpea-derived fibers enhance the formation of the *B. subtilis* biofilms and the production of the antimicrobial pigment pulcherrimin. Furthermore, electron micrograph imaging confirms the bacterial embedding onto the chickpea fibers, which may provide a survival tactic to shield and protect the bacterial population from environmental insults. Overall, it is believed that chickpea-derived prebiotic substances provide a staple basis for developing functional probiotics and synbiotic food.

## 1. Introduction

The healthier dietary choices include an enhancement in the consumption of plant-based food such as whole grains and legumes, which reduces blood lipid levels, postprandial glucose, and insulin levels [1,2,3]. These foods are rich in prebiotic substances that are beneficial to probiotic microorganisms residing in the host gut. Probiotics have been defined as “a live microbial food ingredient that is beneficial to health and affects the host through its effects in the intestinal tract” [4]. In the right amounts, it confers health benefits to the host organism. These benefits include restoring or augmenting the healthy microflora [5], inhibiting the growth of pathogenic bacteria [6], and releasing by-products that the host organism can use for metabolic activities [7]. Most of the probiotic species are often sourced from dairy foods [8].

Prebiotics are the non-digestible components of our diet serving as a stimulant for growing certain bacteria, preferably probiotic species, in the gastro-intestinal tract [9]. They appear to be vital to developing the proper intestinal microflora [8], since are required for feeding probiotic microorganisms in the gastrointestinal tract [2]. The breakdown of these substances yields products that benefit the gut and help distant organs when released into the blood circulation [7]. Short-chain fatty acids (SCFAs), a fermentation by-product, lower colon pH [10]; more so, the SCFAs may trigger the establishment of probiotic *Bacilli* through inducing biofilm formation pathway [11]. Moreover, peptidoglycans released from the fermentation of certain prebiotics aid the innate immune system in fighting pathogenic bacteria [7].

Chickpea, *Cicer arientinum* L., is a legume indigenous to the Middle East and used in making plant-based milk, hummus, and falafel, among other delicacies [12]. Chickpea is rich in micronutrients, minerals, and dietary fibers [13]. Production of chickpea milk is accomplished by crushing the legumes with a significant amount of water. After the extraction of proteins and starch from the residue, often the remainder is discarded or used as animal feed. The discarded part appears to be very rich in polysaccharides including dietary prebiotics.

Probiotic *Bacilli* often form multicellular communities called biofilms, which help them to survive harsh environmental conditions. Biofilm formation has previously been investigated mainly for its negative impact in various industries [14,15,16], but beneficial biofilm concept has recently regained interest, for instance, due to their ability to ensure the safe delivery of probiotics to the lower gastrointestinal tract [17,18]. Since chickpea polysaccharides have been shown to facilitate certain physiological processes of probiotic *Bacilli* [19], we hypothesized that they could trigger signaling pathways for biofilm formation. In *B. subtilis*, the model organism for Gram-positive *Bacilli*, the biofilm formation is regulated by the KinD-Spo0A pathway [20]. According to this mechanism, the accumulation of phosphorylated Spo0A-P leads to upregulation of major matrix operons *epsA-O* and *tapA-sipW-tasA*. In addition, the activated Spo0A results in the generation of pulcherriminic acid via the *yvmC-cypX* operon [21].

In this study, we report that the chickpea-derived dietary fibers trigger notably biofilm formation. Moreover, these prebiotic substances significantly increase the production of brownish pigment associated with the synthesis of pulcherrimin molecules by *B. subtilis*. We further show that the cells harbored in a self-produced biofilm matrix could notably improve the survivability of *B. subtilis* cells through a simulated digestion system.

## 2. Materials and Methods

### 2.1. Strains and Growth Conditions

Bacterial strains used in this study and their origins are listed in Appendix A. Wild-type (WT) *B. subtilis*, NCIB 3610, and its derivatives were typically grown in Lysogeny broth (LB) (Difco, Franklin Lakes, NJ, USA) comprising 10 g of tryptone, 5 g of yeast extract, and 5 g of NaCl (per liter) or on LB broth solidified with 1.5% agar (Difco, Franklin Lakes, NJ, USA). Prior to making starter cultures, *B. subtilis* cells were grown on agar solidified plates overnight at 37 °C. Starter culture of *B. subtilis* cells was prepared using a single bacterial colony in 5 mL of LB broth and incubated at 23 °C with shaking at 150 rpm overnight.

#### 2.1.1. Chickpea Milk Preparation

Chickpea Milk (CPM) was prepared as reported by Rajasekharan et al. [19]. Briefly, 93.8 g of Kabuli-type chickpea (*Cicer arietinum*, L.) seeds were soaked in 600 mL of double-distilled water (DDW) and incubated for 12 h at room temperature; afterward, fresh DDW was used to replace the old one, and the seeds were crushed using a Waring commercial blender. The liquid chickpea suspension was filtered using a cheesecloth to obtain chickpea milk (CPM), which was boiled at 90 °C for 10 min. Finally, the generated CPM was autoclaved and stored for further assays.

#### 2.1.2. Preparation of the Prebiotic Stock Suspensions

All media used as treatments in this study and their origins are listed in Appendix A. For this study, 10 g of either chickpea fiber (CPF), Benefiber powder (WF), or cellulose fiber (CF) were weighed into a reagent bottle supplemented with 100 mL DDW to generate a 10% *w/v* stock solution. The sterilized suspensions were stored and used for further assays.

### 2.2. Macroscopic Assessment of Biofilm Formation in the Prebiotic Enriched Media

The starter culture of *B. subtilis* was prepared as described above. LB broth was supplemented with different concentrations of the above-mentioned fibers. For pellicle formation assays, 5 μL of the starter culture was pipetted into media containing either 4 mL of LB, LB + CPF, LB + WF, or LB + CF into 12-well polystyrene plates. They were incubated at 30 °C supplemented with slight shaking (30 rpm for 72 h); the generated biofilms were visualized using either a regular camera (iPhone 11) or Nikon fluorescent microscope (Nikon Eclipse Ti2, Tokyo, Japan).

### 2.3. Growth Curve Analysis of B. subtilis in the Presence of Different Prebiotic Fibers

For starter cultures, the WT and matrix mutants (∆*eps* and ∆*tasA*) were prepared as initially described. The strains were thereafter inoculated in either LB or LB supplemented with 1% *v/v* of dietary fiber. Each sample was then harvested at 2 h intervals and serially diluted in 900 µL phosphate-buffered saline (PBS). Viable cell counts were determined using the CFU counting on LB agar plates. Log CFU/mL was determined, and a graph was plotted against time.

### 2.4. Determining the Effect of Prebiotic Supplementation on the Expression of Biofilm Matrix Operon, tapA, Using β-Galactosidase Assay

To investigate the effect of dietary fiber on matrix gene expression, *B. subtilis* harboring the transcriptional fusion of *tapA* promoter with a gene encoding for an enzyme beta-galactosidase (YC 121) was inoculated into 5 mL of fresh LB and incubated overnight at 23 °C shaking at 150 rpm. Afterward, 4 μL of the starter suspension was inoculated into either 4 mL LB or the fiber-enriched LB in the 12-well polystyrene plates. The plates were incubated overnight for 16 h at 30 °C with slight shaking (of 30 rpm). The optical density of the cell samples was normalized using OD_600_. One milliliter of cell suspensions was collected and assayed for *β*-Galactosidase activity as described previously by Thibodeau et al. [22].

### 2.5. Visualizing the Impact of Prebiotics on Biofilm Formation Using Scanning Electron Microscopy

The cells of *B. subtilis* NCIB 3610 and its derivative mutants were prepared as described above. Four microliters of each strain were inoculated into a 4 mL LB medium supplemented with either 1% or 3% (*v/v*) prebiotic substances. One milliliter of suspension from each setup was centrifuged for 2 min and the pellets were washed with DDW twice to remove the spent medium. Then, 5 µL of pellets from each sample was placed on polylysine-coated glass slides and left open in the biological hood for 12 h to dry. Prior to analysis in the SEM, the slides were coated with gold/palladium coating (20:80), at 12 mA voltage and 1 nm thickness.

### 2.6. Analysis of Survival of B. subtilis Cells Transitioning through In Vitro System

To determine the survivability of *B. subtilis*, 50 μL of bacterial suspension were introduced into 5 mL of growth medium (either LB or LB supplemented by one of the prebiotic substances) and incubated at 37 °C shaking at 100 rpm. In vitro digestion was carried out similarly as reported previously [18].

### 2.7. Determining the Effect of CPF on the Generation of Pulcherrimin by B. subtilis

The WT cells of *B. subtilis* were introduced into either CPM or CPM enriched with CPF in a conical flask. The cells were incubated for 48 h, and the subsequent extraction of pulcherrimin was carried out similarly to Rajasekharan et al. [19].

### 2.8. Statistical Analysis

The numerical data obtained were analyzed statistically by means of the analysis of variance following a post hoc t-test at a significance level of *p* < 0.05, to compare the control and tested samples. The results are based on three biological repeats performed.

## 3. Results

### 3.1. Chickpea Fibers Enhance Pellicle Formation and Pulcherrimin Production by Bacillus subtilis

The richness of chickpea milk (CPM) in resistant starch fibers, as well as in iron (Fe^3+^), appears to enable proliferation and production of the antimicrobial pigment pulcherrimin by *B. subtilis* [19]. In the current study, we found that enriching the CPM with chickpea-derived prebiotic substances results in the formation of an improved pellicle (a biofilm at air–liquid interface) (Figure 1a). Furthermore, the supplemented CPM showed an accumulation of reddish-pink pigmentation (Figure 1a), which was identified previously as pulcherrimin [19]. At higher concentrations of chickpea fiber (CPF) (either 3 or 5%), pellicle formation and pulcherrimin production was enhanced (Figure 1), while at 1% of CPF concentrations, there was no significant difference, compared with CPM. We further purified the pulcherrimin and estimated its total yield, which was found as highest in the presence of 5% of CPF (Figure 1b).

A growth curve analysis was performed to ascertain whether the observed induction of biofilm formation by *B. subtilis* cells was determined by an increase in cell biomass in the presence of the prebiotic substances. Evidently, the cells grew well in all the tested media, with CPF having slightly higher cell counts (Figure 1c). However, there was no significant difference between the growth rates of bacterial cells in the tested media. Nonetheless, as shown in Figure 1d, the fluorescently tagged *B. subtilis* cells (YC161) grown in the presence of different doses of CPF successfully colonized chickpea fibers, characterized by producing a certain type of autofluorescence [19].

Next, the biofilm induction assays were conducted (Figure 2) on Lysogeny broth (LB), a laboratory medium that does not support *B. subtilis* biofilm formation. In addition to CPF, other plant-based fibers (CF and WF) were also tested. CPF (3%) induced robust pellicles in wild-type *B. subtilis*, accompanied by pulcherrimin production. CF induced fragile pellicles without pigmentation, while WF could not trigger any notable pellicle formation. Furthermore, *B. subtilis* matrix mutants were unable to form pellicles in the presence of all the tested fibers, but the pigment production was evident in mutants treated with CPF (3%). These findings suggest that among the tested fibers, CPF is the only fiber rich in iron (Fe^3+^), which coordinates the production of pulcherrimin in *B. subtilis* in LB media.

### 3.2. Microscopic Characterization of the CPF-Triggered Biofilm Formation

Scanning electron microscopy was used to visualize the morphology of *B. subtilis* cells grown in a medium supplemented with CPF and other prebiotic substances. The supplementation of various prebiotic substances differently affected the morphological traits of the *B. subtilis* population (Figure 3a). In the control sample, the cells were seen in separated clusters; hence, no notable bundling was observed. A minimal bundling occurred in media enriched with WF, while CPF promoted tight interactions of *B. subtilis*. The matrix mutants showed some film-like sheets formation even though after 72 h of incubation, in the presence of CPF, there were no visible fiber attachments (Figure 2). The CPF-treated samples showed the most convincing and successful bacterial adhesion among the tested prebiotic fibers.

### 3.3. Regulation of Matrix Gene Expression in the Presence of Prebiotic Substances

It was further hypothesized that the chickpea-derived fibers could promote biofilm formation via upregulating the matrix genes expression. Consequently, *B. subtilis* cells were grown in the presence of prebiotic substances to determine the expression of one of the major matrix operons—*tapA* using β-Galactosidase assay [22,23]. It was found that CPF, as well as cellulose fibers, could significantly induce *tapA* expression in a dose-dependent manner (Figure 3b). This implies that prebiotic substances, such as CPF and cellulose, may trigger biofilm formation via activation *tapA* operon as one of the major determinants of biofilm phenotype in *B. subtilis.*

### 3.4. Adherence of B. subtilis to CPF Fibers Governs the Survivability of B. subtilis during In Vitro Digestion

Attachment of *B. subtilis* to prebiotic substances can improve the survivability of cells as they move through strenuous food processing techniques and during food consumption and digestion. Accordingly, *B. subtilis* cells grown in the presence of different prebiotic substances were compared in their survivability during the transition in simulated gut conditions using the experimental system described recently [17].

The survivability of *B. subtilis* cells was significantly increased by 2 log CFU/mL following growth in a medium enriched with CPF, compared with the control sample; whereas the survivability of *B. subtilis* cells increased by about 1 log CFU/mL in the presence of CF and 0.6 log CFU/mL in WF, respectively. These findings indicate that the biofilm formation triggered by CPF could provide optimum protection for *B. subtilis* cells in in vitro digestion conditions. The survivability of *B. subtilis* cells appears to be notably lower in the CF- or WF-treated groups, which could be explained by the modest adherence of the bacterial cells onto those fibers.

## 4. Discussion

The richness in micro-and macronutrients composition of chickpea legumes, including proteins, carbohydrates, and lipids, may provide an excellent medium for the growth of microorganisms [5,12,13]. The legume is often used to produce chickpea milk (CPM), a plant-based alternative for milk of mammalian origin. Furthermore, CPM provides a favorable environment for biofilm formation by probiotic *Bacilli*, particularly *B. subtilis* [19]. After the industrial extraction of proteins and starch from the milk residue, the remainder, which is rich in dietary prebiotics and minerals [13], is often discarded. Therefore, it is conceivable that the chickpea-derived prebiotic substances may notably upgrade different food products as an additive. Moreover, enrichment of those products with probiotic bacteria could propose a staple basis for developing synbiotic foods.

In this study, the CPF supplementation triggered biofilm formation likely via increasing the available micro- and macronutrients, although it is also possible that the CPF may have created a favorable osmotic pressure that signals bacterial cells to opt the biofilm formation pathway [24,25]. In addition, induction in biofilm formation resulted in significant upregulation of *tapA* operon was exclusively related to the CPF. However, other types of tested fibers could just somewhat trigger upregulation in *tapA* expression, especially CF. consequently, we speculate that the availability of certain nutrients in combination with an osmotic pressure may account for the upregulated expression of matrix genes towards biofilm formation.

The CPF-induced biofilm formation and the generation of pulcherrimin by *B. subtilis* may provide various implications for the nutrients and food industry. Microbial cells produce the pulcherrimin molecule, which has been found to have antimicrobial properties against pathogenic bacteria [19,20]. Our data demonstrate that CPF significantly increases the synthesis of this brownish-pink pigment in a dose-dependent manner (Figure 1b). It is believed that CPF could induce the expression of *ymvC* and *cypX* genes, responsible for the synthesis of the pulcherriminic acid. Secondly, because CPF is a rich source of minerals such as iron (required for the conversion of pulcherriminic acid to pulcherrimin), it compels the conversion of the pulcherriminic acid synthesized to pulcherrimin molecule [19]. These properties, coupled with its antimicrobial activity, make it a suitable food additive pigment and improve the iron content of the enriched food.

Keeping in mind another possible application of the CPF-induced biofilm-inspired encapsulation, we assessed the survivability of *B. subtilis* cells during exposure to environmental insults such as acidic pH stress. The biofilm-encapsulated *B. subtilis* cells were found as 100-fold more protected during in vitro digestion. This is consistent with the current belief that biofilms confer protection from environmental insults such as fluctuations in pH levels [17,18,26,27]. The ability of CPF to trigger this phenomenon makes it an optimum medium for bioencapsulation of the probiotic cells [17].

## 5. Conclusions

Taken together, the ability of the CPF fibers to trigger biofilm formation by activation of *tapA* operon and the induced synthesis of pulcherrimin molecule provide novel avenues for developing functional food products. The utmost application of the conceptual idea could be in improving bioencapsulation to ensure the safe delivery of probiotics to the lower gastrointestinal tract (GIT). Finally, upscaling the CPF-induced system for extraction and purification of pulcherrimin could generate a more natural alternative additive for many food products and ensure consumer safety.

## Figures and Tables

**Figure 1 nutrients-13-04228-f001:**
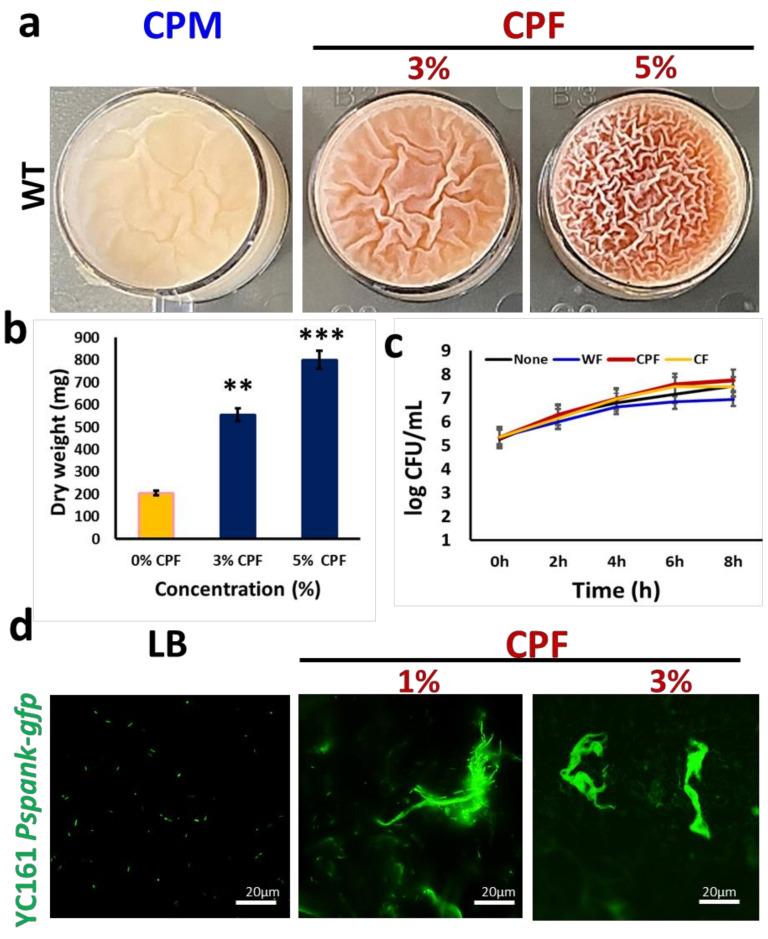
Induction in biofilm formation by *B. subtilis* in response to chickpea fiber (CPF): (**a**) pellicle formation by WT *B. subtilis* in chickpea milk (CPM) with or without supplementation of either 3 or 5% of CPF; (**b**) quantification of pulcherrimin production by *B. subtilis* in the presence of either 3 or 5% of CPF. The graph shows the means ± SEMs of three measurements. ** *p <* 0.01 and *** *p <* 0.001 vs. the non-treated controls; (**c**) growth curve analysis of WT *B. subtilis* during incubation in the fiber’s enriched media. Cells grown for 8 h in LB supplemented with different prebiotic substances were sampled every 2 h to determine the viable cell counts; (**d**) interaction of fluorescently tagged *B. subtilis* cells (YC161, harboring the green fluorescence protein as a reporter) with CPF (1 or 3%) during growth for 16 h in LB medium.

**Figure 2 nutrients-13-04228-f002:**
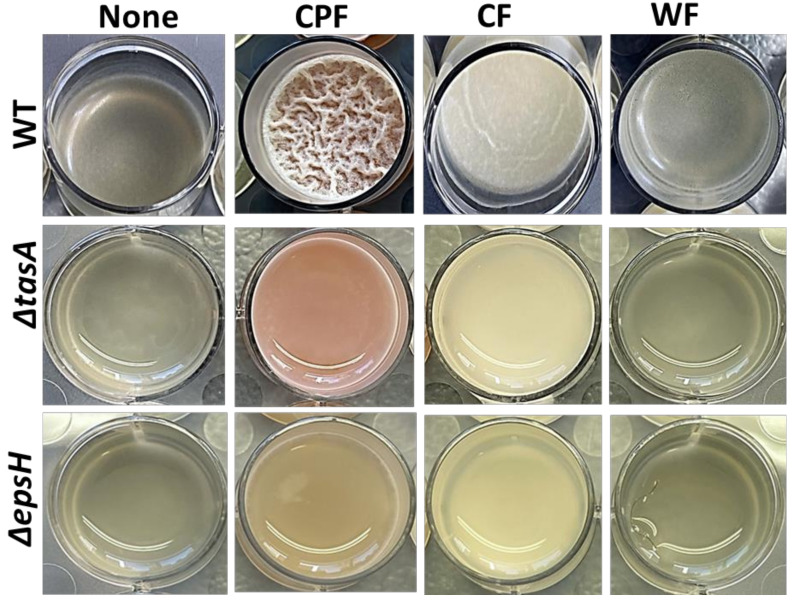
Differential induction of biofilm formation by plant-based prebiotic substances. The 3% (*w/v*) of each prebiotic substance were added to LB and the cells of *B. subtilis* (WT, ∆*epsH*, and ∆*tasA*) were grown in the medium for 72 h prior to capture by iPhone camera. This image is representative of three biological repeats.

**Figure 3 nutrients-13-04228-f003:**
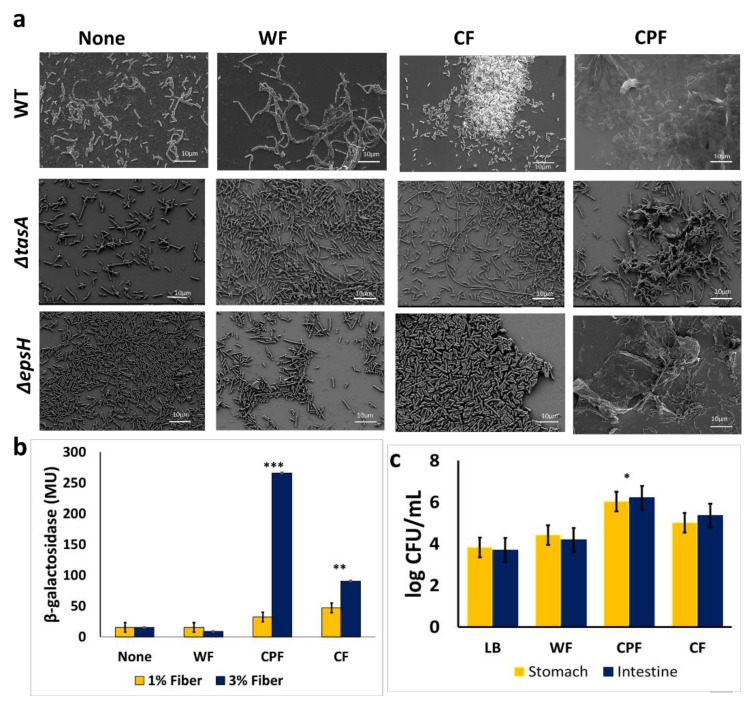
(**a**) An electron micrograph of *B. subtilis* cells (WT, Δ*tasA*, and Δ*epsH*) following 16 h of growth in LB (unsupplemented control) and LB supplemented with different prebiotic substances (WF, CPF, or CF). Images were taken at 1500× magnification. Scale bar: 10 µm; (**b**) measurement of *tapA* expression in the presence of prebiotic substances; *B. subtilis* cells harboring the transcriptional fusion of *tapA* promoter with a gene encoding for an enzyme β-Galactosidase (YC 121) were grown for 16 h in either LB or LB supplemented with the different prebiotic substances. The activity of the *lacZ* gene was assessed in Miller’s units. The graph shows the means ± SEMs of three measurements. ** *p* < 0.01 and *** *p* < 0.001 vs. the non-treated controls; (**c**) survivability of *B. subtilis* grown in different media during in vitro digestion. *B. subtilis* cells were incubated in LB medium supplemented with 1% (*w/v*) dietary fiber for 24 h after which it was further incubated at 37 °C with enzymes, pH modification like the conditions in the stomach. Unsupplemented LB served as a control. The graph shows the means ± SEMs of three measurements. * *p* < 0.05 vs. the non-treated controls.

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
