# Peer review of "Chickpea-Derived Prebiotic Substances Trigger Biofilm Formation by Bacillus subtilis"

_nutrients, 2021, doi:10.3390/nu13124228_

Round 1
Reviewer 1 Report
My comments/questions are as follows:
Title: Chickpea derived prebiotic substances trigger biofilm formation by Bacillus subtilis
The topic looks interesting however, I would like to know that how this research is different from previously published article as Rajasekharan, S.K.; Paz-Aviram, T.; Galili, S.; Berkovich, Z.; Reifen, R.; Shemesh, M. Biofilm Formation onto Starch Fibres 350 by Bacillus subtilis Governs Its Successful Adaptation to Chickpea Milk. Microb. Biotechnol. 2021, 14 (4), 1839–1846. doi: 351 10.1111/1751-7915.13665 https://doi.org/10.1111/1751-7915.13665.
The authors have taken very casual approach in writing the manuscript
- Spelling needs to be checked properly
- Abbreviation needs to be expanded at first place
- Results are not described properly in the text and need not to discussed in the results section by citing the reference as there is separately discussion section where authors need to discuss the results. Authors needs to clearly describe only what are observations has been observed from the experiments undertaken.
- Biochemical test has not been performed after inoculation of Bacillus subtilis so as to know the purity of biofilms and also molecular characterization of biofilms would have been carried out for better understandings.
- In reference section lots of non-uniformities are observed and not followed the journal guidelines in writing references.
- Space between number and units are not uniformly written correct for example somewhere written like 5 g and somewhere mentioned as 5mL so needs to be corrected.
Specific comments are as follows:
- In line no.14 check space before prebiotic
- In line no. 55 GIT can be expanded at first place
- In materials and methods section of line no 70 mentioned Table S1 however, I was unable to find out the table.
- What is WT mentioned in line no. 71
- °C is not written correctly at many places
- In line no 77 LAB & MRS need to be expanded
- In 2.1.1 of line no 80 mentioned as Chickpea Milk (CPM) was made as reported by [19] but it could have been written as Chickpea Milk (CPM) was made as reported by xyz et al. [19]
- In line no. 85 mentioned that CPM was boiled for 10 min at what temperature?
- In line no. 85 full stop is missing at the end of sentence.
- In line no. 92 instead of ‘The’ can be written as ‘the’.
Author Response
The topic looks interesting however, I would like to know that how this research is different from the previously published article as Rajasekharan, S.K.; Paz-Aviram, T.; Galili, S.; Berkovich, Z.; Reifen, R.; Shemesh, M. Biofilm Formation onto Starch Fibers by Bacillus subtilis Governs Its Successful Adaptation to Chickpea Milk. Microb. Biotechnol. 2021, 14 (4), 1839–1846. doi: 351 10.1111/1751-7915.13665 https://doi.org/10.1111/1751-7915.13665.
Answer: We aimed in the study to evaluate the contribution of the extracted dietary fibers on the physiological traits of bacteria, which appears to be a follow-up research of our previous article mentioned by the reviewer. Thus, the current study demonstrates explicitly which dietary fiber could trigger the biofilm formation and the pigment production most profoundly. We have accordingly explained in the text about the novelty and rationale of the study as well as the experiments performed using the purified fibers. In the supplementary data, a section describing the extraction of these fibers from Chickpea grains buttresses was also added.
The authors have taken a very casual approach in writing the manuscript
- Spelling needs to be checked properly
- Abbreviation needs to be expanded at the first place
Results are not described properly in the text and need not to discussed in the results section by citing the reference as there is separately discussion section where authors need to discuss the results. Authors needs to clearly describe only what are observations has been observed from the experiments undertaken.
Answer: We have carefully rewritten the text to avoid confusion and added references in the appropriate places.
- Biochemical test has not been performed after inoculation of Bacillus subtilis so as to know the purity of biofilms and also molecular characterization of biofilms would have been carried out for better understandings.
Answer: In addition to the pure B. subtilis strain used in the experiment, we evaluated the morphological appearance of the bacteria (as microscopically as well as microscopically) to confirm the purity of the strains during biofilm formation. Furthermore, the reporter genes tagged cells were used to characterize them during biofilm formation.
- In the reference section lots of non-uniformities are observed and not followed the journal guidelines in writing references.
Answer: This issue has been fixed.
- Space between number and units are not uniformly written correct for example somewhere written like 5 g and somewhere mentioned as 5mL so needs to be corrected.
Answer: This point has been addressed.
Specific comments are as follows:
- In line no.14 check space before prebiotic
Answer: it has been corrected.
- In line no. 55 GIT can be expanded at first place
Answer: This point has been fixed.
- In materials and methods section of line no 70 mentioned Table S1 however, I was unable to find out the table.
Answer: The TableS1 is now provided in the supplementary material.
- What is WT mentioned in line no. 71
Answer: It is the abbreviation for the word “wild type' and is now clearly indicated.
- °C is not written correctly in many places
Answer: this point has been fixed.
- In line no 77 LAB & MRS need to be expanded
Answer: It has been deleted because it had no relevance to this article.
- In 2.1.1 of line no 80 mentioned as Chickpea Milk (CPM) was made as reported by [19] but it could have been written as Chickpea Milk (CPM) was made as reported by xyz et al. [19]
Answer: it has been corrected.
- In line no. 85 mentioned that CPM was boiled for 10 min at what temperature?
Answer: it is now indicated in the text.
- In line no. 85 full stop is missing at the end of the sentence.
Answer: This point has been addressed.
- In line no. 92 instead of ‘The’ can be written as ‘the’.
Answer: it has been corrected.
Reviewer 2 Report
In this Study the Authors analyzed the activity of chickpea derived prebiotic substances towards beneficial bacteria like B. subtilis.
The work is original and well-done.
Nevertheless, I would encourage the Authors to review the English scientific language they used in the Manuscript. There are a number of sentences which do not make sense; in addition wrong tens and terms have been used. Several grammar mistakes are present along the Text
Check the title: two different titles are shown on top and bottom coverpage
Rephrase the “Abstract” paragraph
Check the reference list. Some points lack appropriate formatting
Author Response
In this study, the Authors analyzed the activity of chickpea-derived prebiotic substances towards beneficial bacteria like B. subtilis.
The work is original and well-done.
Nevertheless, I would encourage the Authors to review the English scientific language they used in the Manuscript. There are several sentences that do not make sense; in addition, wrong tens and terms have been used. Several grammar mistakes are present along with the Text
Answer: We thank the reviewer for positively evaluating our study. As suggested, we have carefully rewritten the text to avoid confusion or grammar issues.
Check the title: two different titles are shown on the top and bottom coverage
Answer: This point has been addressed.
Rephrase the “Abstract” paragraph
Answer: The abstract has been rewritten as suggested.
Check the reference list. Some points lack appropriate formatting
Answer: The point has been addressed.
Round 2
Reviewer 1 Report
It has been improved and needs minor correction.

Author Response
We would like to thank the reviewer for the comment and corrections. We found them as useful, which contributed to improve the manuscript notably.
Sincerely,
Moshe Shemesh
